# SI-BiViT: Binarizing Vision Transformers with Spatial Interaction

## ABSTRACT

Binarized Vision Transformers (BiViTs) aim to facilitate the efficient and lightweight utilization of Vision Transformers (ViTs) on devices with limited computational resources. Yet, the current approach to binarizing ViT leads to a substantial performance decrease compared to the full-precision model, posing obstacles to practical deployment. By empirical study, we reveal that spatial interaction (SI) is a critical factor that impacts performance due to lack of token-level correlation, but previous work ignores this factor. To this end, we design a ViT binarization approach dubbed SI-BiViT to incorporate spatial interaction in the binarization process. Specifically, an SI module is placed alongside the Multi-Layer Perceptron (MLP) module to formulate the dual-branch structure. This structure not only leverages knowledge from pre-trained ViTs by distilling over the original MLP, but also enhances spatial interaction via the introduced SI module. Correspondingly, we design a decoupled training strategy to train these two branches more effectively. Importantly, our SI-BiViT is orthogonal to existing Binarized ViTs approaches and can be directly plugged. Extensive experiments demonstrate the strong flexibility and effectiveness of SI-BiViT by plugging our method into four classic ViT backbones in supporting three downstream tasks, including classification, detection, and segmentation. In particular, SI-BiViT enhances the classification performance of binarized ViTs by an average of 10.52% in Top-1 accuracy compared to the previous state-of-the-art. The code will be made publicly available.

## CCS CONCEPTS

• **Computing methodologies** → *Computer vision problems*; • **Networks** → Network design principles.

## KEYWORDS

Model Binarization; Vision Transformer; Spatial Interaction;Plug-and-Play

## 1 INTRODUCTION

Recently, Vision Transformers (ViTs) [14] have achieved significant success across various tasks, including classification [6, 14, 48], object detection [15, 34, 51, 62] and segmentation [17, 60]. Meanwhile, the increasing model size and computational demands have limited their deployment in mobile and Internet of Things (IoT) devices. To address these efficiency bottlenecks, model compression

**Unpublished working draft. Not for distribution.**

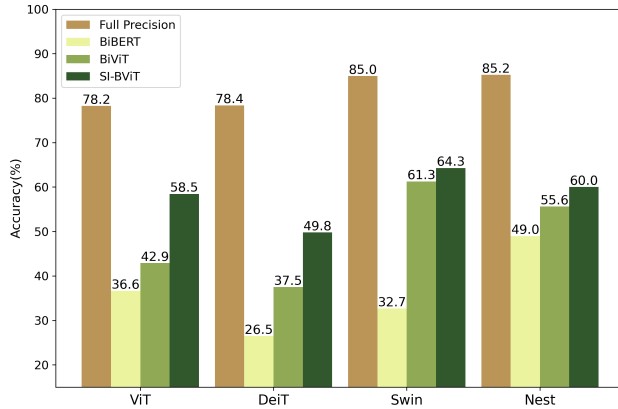

**Figure 1: Performance comparison between our SI-BiViT and other binarized vision transformer approaches. SI-BiViT surpasses the state-of-the-art in all ViT backbones.**

techniques such as distillation [22, 47], pruning [41, 63] and quantization [13, 32, 33, 59] have been extensively explored. Among them, Binarized Neural Network (BNN) [36, 44, 45, 55, 61] aggressively quantizes weights and activations to 1-bit and utilizes efficient XNOR and popcount bit-wise operations to accelerate inference speed as well as reduce energy consumption significantly. Binarized neural networks can typically achieve up to 32 times memory saving and 58 times speedup [45].

Although model binarization yields efficiency gains, it severely compromises performance due to substantial information loss. Several studies have aimed to bridge the performance gap between binarized models and their full-precision counterparts. Qin et al. [43] and He et al. [19] investigate the optimal threshold of attention score to preserve more information after binarization. Some studies [26, 35, 43] focus on optimization in binarized vision transformers by utilizing distillation and preserving more knowledge from pre-trained models. While existing approaches have demonstrated some effectiveness, a substantial performance gap persists between binarized ViTs and their full-precision counterparts, as evidenced by an average of 30.17% in ViT in Figure 1.

For the first time, we empirically demonstrate that the lack of token-level correlation severely limits the performance of binarized ViTs. We introduce the concept of Spatial Interaction Capability (SIC) to quantify the contribution of different modules to spatial interaction. Our analysis reveals that the MLP module solely influences feature extraction in each token independently, resulting in a spatial interaction capability of zero, which significantly hinders spatial interaction in binarized ViTs. To further investigate the relationship between the model's performance and spatial interaction, we progressively increase spatial interaction capability across four ViT backbones. As illustrated in Figure 2, the performance exhibits a notable increase with more spatial interaction, indicating a positive correlation between spatial interaction and performance. These findings indicate that existing binarized vision transformers

indeed lack spatial interaction and suffer considerable performance degradation.

In this paper, we introduce a ViT binarization approach named SI-BiViT, which integrates spatial interaction into the binarization process. Specifically, we propose a dual-branch structure by incorporating a Spatial Interaction Module(SIM) alongside the existing MLP module. This design not only utilizes knowledge from pre-trained ViTs through distillation over the original MLP but also enhances spatial interaction through the SI module. To effectively train these two branches, we devise a two-stage decoupled training strategy. In the first stage, we optimize the MLP branch by distilling from pre-trained models, enabling the MLP to retain more information through binarization. However, due to the significant differences between the SI module and the MLP module, incorporating the SI module at this stage may lead to conflicting gradient problems. Hence, we defer the addition of our SI module to the second stage, where we optimize the model solely by supervision of classification loss without distillation. In the second stage, we introduce our SI module to form the dual-branch structure. To mitigate the increase in computational and memory costs resulting from the dual-branch structure, we utilize a compact token-mixing linear as our SI module. Furthermore, we introduce a channel-wise balancing factor to dynamically adjust the magnitude of the SI module.

It is important to note that SI-BiViT is plug-and-play in two aspects. Firstly, it can directly replace the vanilla MLP modules and generalize to various ViT backbones, including Deit[48], Swin[34], Nest[62]. Secondly, SI-BiViT is orthogonal to existing Binarized ViTs approaches [19, 43] and can be further combined to improve their performance. Extensive experiments on various ViT backbones and tasks demonstrate the strong flexibility and effectiveness of SI-BiViT. As illustrated in Figure 1, SI-BiViT increases the classification performance of binarized ViTs by an average of 10.52% in Top-1, and increases detection and segmentation tasks by an average of 1.1% in AP.

In summary, our contributions are three-fold:

- We are the first to study spatial interaction in binarized vision transformers. Our empirical findings demonstrate that existing binarized VITs lack spatial interaction due to a lack of token-level correlation, which is a critical factor that affects performance.
- We propose SI-BiViT to incorporate spatial interaction in the binarized ViTs. Concretely, we introduce a plug-and-play dual branch that not only leverages knowledge from pretrained ViTs by distilling over the original MLP module, but also enhances spatial interaction via the introduced SI module. Additionally, we design a decoupled training strategy to effectively optimize the dual branch.
- Extensive experiments on various ViT backbones and downstream tasks demonstrate the strong flexibility and effectiveness of SI-BiViT.

## 2 RELATED WORKS

### 2.1 Vision Transformer

The Transformers [49] is originally proposed to solve natural language processing (NLP) problems and achieves great success by its long-range interaction mechanism. Vision Transformer (ViT)

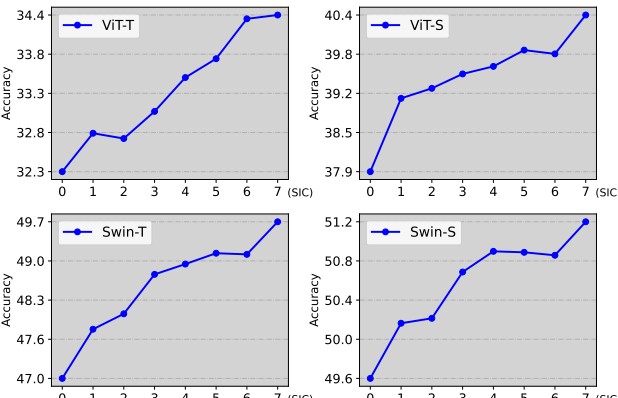

**Figure 2: Performance comparison when the models' spatial interaction capacity(SIC) increases. The performance shows a clear upward trend as SIC increases.**

[14] first introduces transformers into the field of computer vision. Images are divided into non-overlapping patches, with each patch being embedded into a token. These tokens are then processed using the same pipeline as those used in NLP. Deit [48] optimizes data augmentation and distillation tokens for ViT and further enhances the generalization ability of ViT. However, these models suffer from the drawback of high computational complexity, resulting in significantly lower efficiency problems. This is mainly due to the quadratic computational complexity of the self-attention mechanism [23, 50] and the fixed size of embeddings throughout the computation process [58]. To mitigate these issues, LGViT[54] incorporates heterogeneous exiting heads, namely, local perception head and global aggregation head, to achieve an efficiency-accuracy trade-off. Swin Transformer [34] and Nest Transformer [62] design compact and hierarchical architectures to capture global and local information more efficiently. Additionally, other studies explore quantization on ViT[13, 16, 27, 37], pruning techniques [56, 57] and ditillation[9, 29]. Compared to these compression approaches, model binarization shows significant advantages in reducing calculation and memory.

### 2.2 Binarized Neural Network

As a pioneer work in model binarization, BNN[45] quantizes both weights and activations to 1 bit, which significantly accelerates the inference of neural networks and saves memory occupancy through XNOR and popcount operator. However, binarization will lead to a severe performance drop, causing a severe performance drop. Consequently, numerous studies endeavor to enhance model binarization techniques. [38] proposes Bi-Real net, which connects the real activations to activations of the consecutive block through an identity shortcut and further strengthens the network's representation ability. [28] considers the angle alignment between the full-precision weight vector and its binarized version and proposes to rotate the weight vector to reduce the angular bias. These works primarily focus on minimizing quantization errors [3, 28, 31], designing better Straight-Through Estimator (STE) [4, 38], improving better distribution for BNN [12, 36] and increasing the information capacity of binarized models [44].

## 2.3 Binarized Transformer

Binarization of transformer-based models has attracted a lot of interest since they achieve great success in a lot of downstream tasks. Unfortunately, existing binarization approaches designed for convolutional neural networks cannot migrate well to the transformers due to distinct architectural differences. BiBERT [43] proposes a Bi-Attention structure to maximize the information entropy of the self-attention module and further proposes direction-matching distillation to reduce the gradient mismatch problem in attention. Liu et al. [35] propose a two-set binarization scheme and a successively distilling mechanism to preserve more information from pre-trained models. He et al. [19] propose softmax-aware binarization to address the long-tailed distribution of softmax attention. They also introduce a cross-layer binarization scheme to decouple the binarization of the attention module and MLP module. Li et al. [26] propose a ranking-aware distillation method to alleviate attention distortion. However, most existing approaches overlook the key effect of spatial interactions, which severely hinders the performance of binarized ViTs.

## 3 METHOD

### 3.1 Preliminaries

Binarized Neural Networks generally use sign function to quantize weights and activations to -1 and +1 [45]. Unfortunately, the gradient of sign function is non-differentiable and thus cannot be updated via backpropagation. To overcome the non-differentiable issue, Straight-Through Estimator(STE) [2] is applied to approximate gradients in the back-propagation. The formula of forward and backward is:

$$\textbf{Forward:} \quad x^b = \text{Sign}(x) = \begin{cases} +1, \text{if } x \geq 0 \\ -1, \text{otherwise,} \end{cases} \quad (1)$$

$$\textbf{Backward:} \quad \frac{\partial \mathcal{L}}{\partial x^b} \approx \begin{cases} \frac{\partial \mathcal{L}}{\partial x}, & \text{if } |x| \leq 1 \\ 0, & \text{otherwise.} \end{cases} \quad (2)$$

where the $b$ denotes the binarized values. Gradient clipping[20] is commonly used to prevent unstable weight oscillation. STE only backpropagates gradients when the absolute values are smaller than 1 and truncates them when they exceed 1.

In vision transformer, images are represented by token sequences $A \in \mathbb{R}^{n \times d}$, where $n$ is the sequence length and $d$ is the length of embedding. We binarize them as follows:

$$A^b = \text{Sign}(A - \gamma) \quad (3)$$

where $A$ and $A^b$ denote input real-valued activation and binarized activation respectively. $\gamma \in \mathbb{R}^d$ is a channel-wise learnable factor introduced by [36], which can better control the threshold of activations in binarization.

Similarly, the weights $W \in \mathbb{R}^{d \times d'}$ in the linear layer are also binarized channel-wisely. In order to reduce quantization error, a scaling factor $\theta_w$ is commonly adopted[44]. The weights get binarized as follows:

$$W^b = \theta_W \text{ sign}(W - \mu_w), \quad \theta_W = \frac{1}{n} \|W\|_{l1} \quad (4)$$

where $\mu_w$ is the mean value of $W$.

When weights and activations are all binarized, the computationally heavy operations of floating-point matrix multiplication are replaced by binarized matrix multiplication, which can be accelerated by efficient bitwise XNOR operations and popcount operations [3]:

$$\text{Bi-Linear}(A^b, W^b) = \text{popcount}(\text{XNOR}(A^b, W^b)), \quad (5)$$

where $A^b$ and $W^b$ are binarized activations and binarized weights.

### 3.2 Spatial Interaction in ViT

For the first time, we empirically reveal lack of token-level correlation severely limits the performance of binarized ViTs. In vision transformer, an input image is represented as a sequence of patches and each token represents the feature of the patch. Denotes these tokens as $Z \in \mathbb{R}^{n \times d}$, where $n$ is the length of the sequence and each token $Z_i$ is a $d$-dimension vector, describing the feature of a patch. In order to analyze spatial interaction in the ViT, following the definition of representation ability [24, 38], we introduce the concept of **spatial interaction capability (SIC)** to measure token-level correlation in binarized vision transformers:

$$\text{SIC}(Z) = \sum_i^n \text{Inter}(Z_i) \times \text{Range}(Z_i) \quad (6)$$

where $\text{Inter}(Z_i)$ represents the number of tokens $Z_i$ interacted with. $Range(Z_i)$ denotes the range of interaction strength. The multi-head self-attention calculates attention score matrix $attn \in \mathbb{R}^{n \times n}$ and $attn_{i,j}$ represents the interaction strength between token $i$ and token $j$. Thus, each token needs to interact with all other $n - 1$ tokens in MSA. So $\text{Inter}(Z_i) = n - 1$ in MSA. In binarized models, the $Range(Z_i)$ of MSA is 2, because $attn_{i,j} \in \{0, 1\}$. 0 indicates these two tokens have no interaction and 1 indicates there is an interaction. Thus, The spatial interaction capability of MSA in binarized ViT can be represented as $2n(n - 1)$, where $n$ is the length of tokens.

As another fundamental component in ViT, the multi-layer perception(MLP) is stacked by linear layers. The linear layer processes each token $Z_i$ using a weight matrix $W \in \mathbb{R}^{d \times d'}$, thereby extracting information in each token independently and expanding its dimension to $d'$:

$$\text{Linear}(Z, W) = \sum_i^n Z_i W \quad (7)$$

where $Z_i \in \mathbb{R}^{1 \times d}$ is a token from $Z$. It indicates that each $Z_i$ after linear layers has no interaction with other tokens, thus $\text{Inter}(Z_i) = 0$ in MLP module. So the spatial interaction capability of MLP is $n \times 0 \times 2 = 0$. While the SIC of the MSA module is $2n(n-1)$, the MLP module contributes 0 spatial interaction, thus severely hindering the spatial interaction in binarized ViTs.

To further uncover the relationship between the model's performance and spatial interaction, we gradually increase spatial interaction capability in four ViT backbones by adding more and more MSA modules. We reduce the MLP module accordingly to make the model size and computation roughly the same. As shown in Figure 2, The classification accuracy has a clear upward trend with more and more spatial interaction. The curves on four ViT backbones all demonstrate that there is a positive relationship between spatial

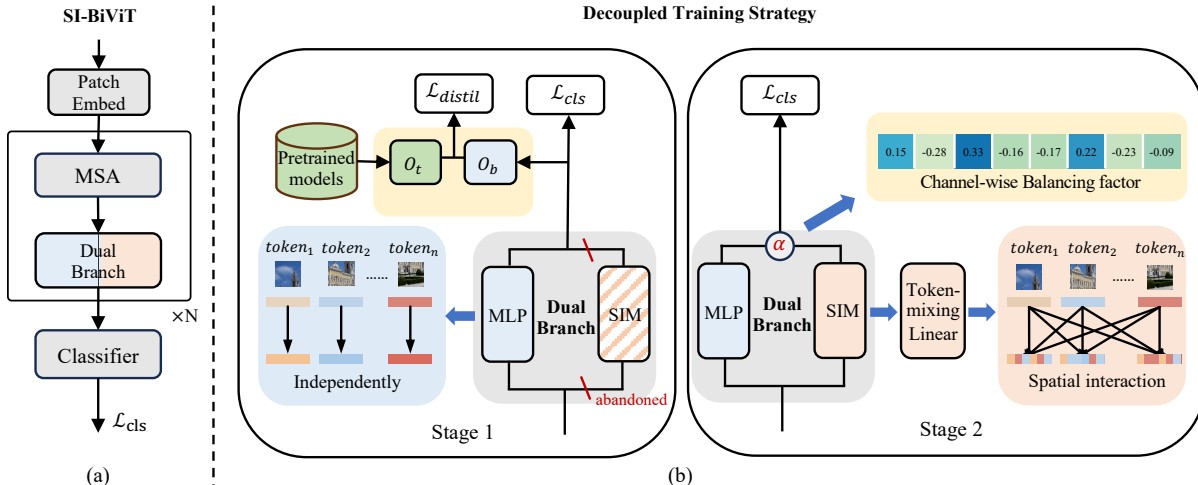

**Figure 3: An overview of proposed SI-BiViT. (a) illustrates the whole architecture of SI-BiViT and we propose a plug-and-play Dual Branch. (b) illustrates the decoupled training strategy of our SI-BiViT. In stage 1, knowledge distillation is applied to preserve information from pre-trained models and SI module is abandoned. In stage 2, we add a token-mixing linear as our SI module to increase spatial interaction and use channel-wise balancing factors to combine the dual branch.**

interaction and performance. This indicates that existing binarized vision transformers do lack spatial interaction and suffer severe performance drops.

## 3.3 SI-BiViT

In this section, we present our SI-BiViT, which integrates spatial interaction into the binarization process in a plug-and-play manner. Specifically, we incorporate an SI module alongside the Multi-Layer Perceptron (MLP) to establish a dual-branch structure. This structure not only capitalizes on the knowledge from pre-trained ViTs by distilling over the original MLP but also enhances spatial interaction through the introduced SI module. Consequently, we devise a decoupled training strategy to effectively train these two branches. The entire framework is depicted in Figure 3.

*3.3.1 Dual-branch structure.* There are various SI modules, such as cycle-FC[8] and multi-head self-attention[49], which can improve token-level correlation. Here, we introduce the Token-mixing Linear[46] which is highly efficient. Denotes these tokens as $Z \in \mathbb{R}^{n \times d}$, where $n$ is the length of the sequence and each token $Z_i$ is a $d$-dimension vector. $Z' \in \mathbb{R}^{d \times n}$ is obtained by transposing $Z$. Token-mixing Linear is represented as:

$$\text{Token-mixing Linear}(Z, W) = \sum_i^d Z_i' W \qquad (8)$$

where $Z_i' \in \mathbb{R}^{1 \times n}$ and $W \in \mathbb{R}^{n \times n'}$ is weight parameters. Token-mixing Linear operation captures spatial information by mixing features from different tokens. According to Equation 6, $Inter(Z_i) = n - 1$ since each token needs to interact with all other tokens. And the spatial interaction ability of Token-mixing Linear is $2n'(n - 1)$. When $n' = n$, its SIC is the same as that of MSA module. Moreover, we can adjust the spatial interaction ability by $n'$. We find it achieves a balance between performance and accuracy when $n' = n$ in Section 4.3.3. Compared to other SI modules[8, 49], token-mixing

**Table 1: The impact of knowledge distillation on different settings. We employ MSA as the single branch here.**

| Module | Distillation | Top-1$_{(\%)}$ |
|---|---|---|
| Single-branch | ✗ | 40.50 |
| Single-branch | ✓ | 38.32$_{(-2.18)}$ |
| Dual-branch | ✗ | 40.95 |
| Dual-branch | ✓ | 46.67$_{(+5.72)}$ |

Linear is highly efficient and outperforms other SI modules in both efficiency and performance.

In model binarization, knowledge distillation is a prevalent and effective technique to improve performance by distilling intermediate features from full-precision models to binarized models. Moreover, distillation can significantly strengthen representation capabilities and relieve coarse gradients.[26, 40, 43]. However, direct applying distillation does not improve performance when we adopt SI module to replace the MLP module As shown in Table 1. This is because there exist substantial differences in architecture between the SI module and MLP module, making it difficult to retain knowledge from pre-trained models. In fact, it may damage the performance after distillation, causing a decrease of 2.18%. To address this issue, we maintain the MLP module and introduce a spatial interaction module alongside it, forming a dual-branch structure. This ensures better distillation as well as improving spatial interaction. There is an obvious improvement(5.72%) in performance as shown in Table 1. Moreover, we reduce the size of the MLP module, as we utilize knowledge distillation, and a smaller MLP can retain most information from pre-trained models without significant performance degradation. Additional details can be found in the Appendix.

*3.3.2 Decoupled training strategy.* We further observe that our SI modules encounter challenges in optimizing due to gradient conflict problems. This refers to the situation where the gradient

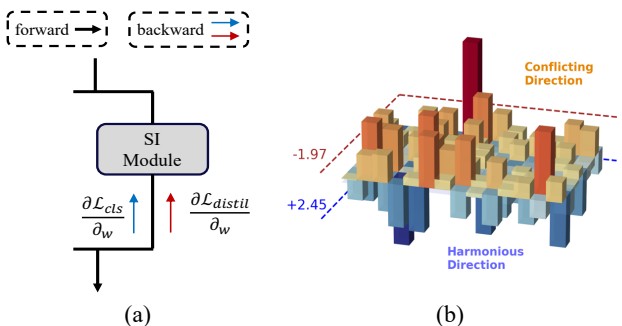

(a)     (b)

**Figure 4: Illustration of gradient conflicting problem. (a) Both the gradient from classification loss and knowledge distillation go into the SI module When knowledge distillation is applied. (b) The up direction indicates the conflicting direction and the down direction indicates the harmonious direction.**

from the classifier does not align with that from knowledge distillation(shown in Figure 4), thereby hindering the optimization of our SI module. We define $W$ to represent the weights of the SI module, $\mathcal{L}_{cls}$ representing the loss from the classifier, and $\mathcal{L}_{distill}$ representing the loss from dense distillation. The distillation loss at one transformer block is formulated as:

$$\mathcal{L}_{distill} = \text{MSE}(A_{student}, A_{teacher}), \qquad (9)$$

where $A_{student}$ is the output after the MLP module in binarized ViTs. $A_{teacher}$ is the corresponding output from pre-trained models. Conflicting gradients are then described as follows:

$$g^+(W) = \sum_i |\frac{\partial \mathcal{L}_{distill}}{\partial W_i}| \odot I_{W_i}$$
$$g^-(W) = \sum_i |\frac{\partial \mathcal{L}_{distill}}{\partial W_i}| \odot -I_{W_i} \qquad (10)$$

where $g^+$ reflects the accumulation of harmonious gradients and $g^-$ indicates the accumulation of conflicting gradients from distillation. $W_i$ is one element from weight of SI module and $\partial \mathcal{L}_{distill}/\partial W_i$ denotes gradient of $W_i$ from knowledge distillation and $\odot$ is Hadamard product. $I_{W_i}$ is a indicator function defined as:

$$I_{W_i} = \begin{cases} 1, & \text{if } \sigma(\frac{\partial \mathcal{L}_{distill}}{\partial W_i}) = \sigma(\frac{\partial \mathcal{L}_{cls}}{\partial W_i}), \\ 0, & \text{otherwise.} \end{cases} \qquad (11)$$

where $\sigma$ is the sign function. $\partial \mathcal{L}_{cls}/\partial W_i$ denotes gradient of $W_i$ from knowledge distillation. As illustrated in Figure 4, $g^-(W)$ reaches 1.97 and it is comparable to $g^+(W)$ at 2.45. This causes optimization oscillation when training the SI module and hinders its optimization.

In order to decouple the training of these two branches, we propose a novel decoupling training strategy that not only leverages knowledge from pre-trained ViTs by distilling over the original MLP, but also enhances spatial interaction via the SI module. in the first stage, we train a model with binarized activations and full-precision weights. At this stage, our dual branch consists only of MLP branch, and we distill the intermediate features after MLP, which preserve most knowledge from the pre-trained model. The SI module is abandoned in the first stage. In the second stage, both the

---

**Algorithm 1** Decoupled Training Strategy.

1: **Input**: input activation:$\mathbf{A}$, weights of MLP: $\mathbf{W^{MLP}}$, weights of SI Module:$\mathbf{W^{SI}}$, channel-wise balancing factor:$\alpha$, weights of teacher MLP: $\mathbf{W_t}$, training epoch of stage-1:$\mathbf{N_1}$, training epoch of stage-2:$\mathbf{N_2}$.
2: **Output**: optimized $\mathbf{W^{MLP}}$ and $\mathbf{W^{SI}}$.
3: **Procedure:**
4:  Calculate $\mathbf{A_b}$ with $\mathbf{A}$ ; $\mathbf{W_b^{MLP}}$ with $\mathbf{W^{MLP}}$ by Equation 3 and 4
5:  Calculate output from MLP $\mathbf{O_b^{MLP}} = \mathbf{W_b^{MLP}A_b}$
6:  **Stage-1:**
7:  For Epoch in $1,2,...,\mathbf{N_1}$:
8:   Calculate output from teacher model $\mathbf{O_t} = \mathbf{W_t A}$
9:   Calculate $\mathcal{L}_{distill}$ between $\mathbf{O_b^{MLP}}$ and $\mathbf{O_t}$ by Equation 9
10:   Calculate $\mathcal{L} = \mathcal{L}_{cls} + \mathcal{L}_{distill}$
11:   Update  $\mathbf{W^{MLP}}$ by $\mathcal{L}$.
12:  End For
13:  **Stage-2:**
14:  For Epoch in $1,2,...,\mathbf{N_2}$:
15:   Calculate $\mathbf{W_b^{SI}}$ with $\mathbf{W^{SI}}$ by Equation 4
16:   Calculate output from SI module $\mathbf{O_b^{SI}} = \mathbf{W_b^{SI}A_b}$
17:   Calculate $\mathbf{O} = \mathbf{O_b^{MLP}} + \alpha * \mathbf{O_b^{SI}}$ by channel-wise balancing factor $\alpha$
18:   Update  $\mathbf{W^{MLP}}$ and $\mathbf{W^{SI}}$ by $\mathcal{L}_{cls}$.
19:  End For
20:  Return optimized $\mathbf{W^{MLP}}$ and $\mathbf{W^{SI}}$.

---

weights and activations are binarized. We do not distill intermediate activations at this stage and inject our spatial interaction module to enhance spatial interaction. Then the conflicting gradient is calibrated and the SI module is only optimized by gradients from the classifier.

To better balance MLP module and SI module, we further introduce a channel-wise self-balancing factor $\alpha \in \mathbb{R}^d$ in the second stage. It can dynamically adjust the magnitude of different channels and be updated in an end-to-end manner with negligible costs. Experiments in Section 4.3.2 show that these factors vary a lot across different channels and do learn unique features.

Overall, the training procedure is illustrated in Algorithm 1.

## 4 EXPERIMENTS

### 4.1 Implementation Details

*4.1.1 Datasets and architectures.* We conduct image classification experiments on two standard benchmarks: TinyImageNet [53] and ImageNet-1k[11]. We also conduct object detection and segmentation tasks on MS-COCO dataset[30]. For classification tasks, data augmentation is adjusted according to DeiT [47], which is common in ViT. To demonstrate the versatility of our approach, we employ four prevail vision transformer backbones: ViT[14], DeiT[48], Swin[34], and NesT[62]. All the layers in Transformer models are binarized except the first input patch-embedding layer and the last classifier layer, which is a common practice of BNNs. The binary operations (BOPs) and floating-point operations (FLOPs) are counted separately, and the operations (OPs) are calculated by OPs = BOPs/64 + FLOPs [36, 45]. For object detection and segmentation tasks, we evaluate the Swin transformer backbone with two classic object detection frameworks Mask R-CNN[18] and Cascade Mask R-CNN[5].

**Table 2: Comparison with state-of-the-art methods on Tiny-ImageNet. "W/A" denotes number of bits in weights and activation.**

| Model | Method | Bits$_{(W/A)}$ | Size$_{(MB)}$ | Bops$_{(G)}$ | Flops$_{(G)}$ | Ops$_{(G)}$ | Top-1$_{(\%)}$ | Top-5$_{(\%)}$ |
|---|---|---|---|---|---|---|---|---|
| ViT-T | Full precision | 32-32 | 22.10 | 0.00 | 1.08 | 1.08 | 77.49 | 91.23 |
| | BiBERT [43] | 1-1 | 0.80 | 1.40 | 0.07 | 0.09 | 23.23 | 48.48 |
| | BiViT [19] | 1-1 | 0.80 | 1.40 | 0.07 | 0.09 | 28.74 | 54.69 |
| | **SI-BiViT** | **1-1** | **0.92** | **1.23** | **0.07** | **0.09** | **46.08** | **71.80** |
| ViT-S | Full precision | 32-32 | 86.67 | 0.00 | 4.25 | 4.25 | 78.25 | 91.22 |
| | BiBERT [43] | 1-1 | 1.65 | 4.90 | 0.13 | 0.21 | 36.65 | 64.35 |
| | BiViT [19] | 1-1 | 1.65 | 4.90 | 0.13 | 0.21 | 42.91 | 69.27 |
| | **SI-BiViT** | **1-1** | **1.84** | **4.20** | **0.15** | **0.21** | **58.45** | **81.04** |
| DeiT-T | Full precision | 32-32 | 22.25 | 0.00 | 1.08 | 1.08 | 78.38 | 93.30 |
| | BiBERT [43] | 1-1 | 0.96 | 1.41 | 0.07 | 0.09 | 26.48 | 52.87 |
| | BiViT [19] | 1-1 | 0.96 | 1.41 | 0.07 | 0.09 | 37.51 | 63.98 |
| | **SI-BiViT** | **1-1** | **1.08** | **1.24** | **0.07** | **0.09** | **49.80** | **75.11** |
| Swin-T | Full precision | 32-32 | 110.60 | 0.00 | 4.37 | 4.37 | 84.97 | 96.18 |
| | BiBERT [43] | 1-1 | 7.03 | 4.44 | 0.30 | 0.37 | 32.69 | 59.83 |
| | BiViT [19] | 1-1 | 7.03 | 4.44 | 0.30 | 0.37 | 61.26 | 82.83 |
| | **SI-BiViT** | **1-1** | **7.41** | **8.04** | **0.32** | **0.45** | **64.29** | **85.16** |
| Nest-T | Full precision | 32-32 | 64.89 | 0.00 | 5.23 | 5.23 | 85.22 | 96.47 |
| | BiBERT [43] | 1-1 | 3.77 | 5.34 | 1.14 | 1.22 | 48.96 | 75.09 |
| | BiViT [19] | 1-1 | 3.77 | 5.34 | 1.14 | 1.22 | 55.63 | 79.51 |
| | **SI-BiViT** | **1-1** | **3.79** | **4.06** | **1.12** | **1.19** | **60.01** | **82.52** |

*4.1.2 Training setup.* For classification tasks, all experiments are implemented with PyTorch [42] and Timm [52] library. For classification tasks, we employ AdamW [39] optimizers with weight decay of 2e-5 and train models for 300 epochs using a cosine annealing schedule with 5 epochs of warm-up. We optimize each stage of the decoupled training strategy with 150 epochs. For object detection and segmentation tasks, we follow the official training pipeline in Swin[34] and replace the backbone with our SI-BiViT. We implement it by mmdetection [7].

*4.1.3 Baseline model.* We first design a binarized vision transformer by adopting several existing techniques. Specifically, these techniques include:

- Binarization method. We use Equation 3 to binarize activation and Equation 4 to binarize weights.
- Activation function. We replace the GeLU activation function with RPRelu introduced by [36]. RPRelu can make activation function adaptively learn the parameters for distributional reshaping.
- Two-stage training. Following [4], in the first stage, parameters are randomly initialized and we only binarize weights. In the second stage, parameters are initialized from stage one and both weights and activation are binarized.
- Normalization. IR-Net[44] finds that BatchNorm[21] can help preserve information and accelerate training. We replace all LayerNorm [1] with BatchNorm [21].

## 4.2 Comparison with State-of-the-art

*4.2.1 Results on Tiny-ImageNet.* As depicted in Table 2, we compare our method with state-of-the-art binarization techniques, such as BiViT [19] and BiBERT [43], across various vision transformer

backbones. We provide performance and efficiency cost metrics for classification tasks on Tiny-ImageNet datasets. Overall, our SI-BiViT achieves superior performance compared to the previous state-of-the-art (SOTA) on all ViT backbones, with an average increase of 10.51%. Specifically, on vanilla ViT backbones, SI-BiViT outperforms the BiViT method by a significant margin of 17.34% in ViT-Tiny and 15.54% in ViT-Small without incurring computation overhead. Additionally, on hierarchical ViT backbones such as the Swin Transformer and NesT Transformer, our SI-BiViT demonstrates an improvement of 3.03% and 4.38%, respectively, showcasing its flexibility and effectiveness. Furthermore, our SI-BiViT effectively narrows the performance gap between binarized vision transformers and their full-precision counterparts. Specifically, we reduce the gap to 19.80% in ViT-Small and 20.68% in Swin-Tiny, thereby enhancing the practicality of binarized ViT models.

*4.2.2 Results on ImageNet.* Moreover, we compare our method with state-of-the-art binarization techniques on ImageNet datasets. We exclude BiBERT [43] from the comparison due to its poor performance, as BiViT [19] consistently outperforms it on all tasks in Tiny-ImageNet dataset. We also utilize larger models such as ViT-base and Deit-small on ImageNet compared to Tiny-ImageNet dataset. We achieve an accuracy of 56.43% in NesT-Tiny and 63.90% in Swin-Tiny, surpassing BiViT by 6.54% and 4.29%, respectively. Notably, our SI-BiViT outperforms BiViT by more than 35% and 21% on ViT-B and Deit-S, nearly doubling the accuracy of BiViT. This highlights the adaptability of our SI-BiViT to more complex classification tasks and larger ViT backbones.

*4.2.3 Results on MS-COCO.* In Table 4, our SI-BiViT outperforms BiViT by 1.6% in object detection and 1.3% in instance segmentation

**Table 3: Comparison with State-of-the-art methods on ImageNet dataset.**

| Model | Method | Bits$_{(W/A)}$ | Size$_{(MB)}$ | Ops$_{(G)}$ | Top-1$_{(\%)}$ |
|---|---|---|---|---|---|
| ViT-S | Full precision | 32-32 | 87.90 | 4.25 | 82.22 |
| | BiViT [19] | 1-1 | 2.88 | 0.21 | 27.73 |
| | **SI-BiViT** | **1-1** | **3.07** | **0.21** | **52.66** |
| ViT-B | Full precision | 32-32 | 345.66 | 16.86 | 84.47 |
| | BiViT [19] | 1-1 | 5.92 | 0.56 | 24.67 |
| | **SI-BiViT** | **1-1** | **6.22** | **0.53** | **60.08** |
| DeiT-S | Full precision | 32-32 | 89.44 | 4.27 | 80.47 |
| | BiViT [19] | 1-1 | 4.42 | 0.21 | 33.99 |
| | **SI-BiViT** | **1-1** | **4.61** | **0.22** | **55.67** |
| Swin-T | Full precision | 32-32 | 113.06 | 4.37 | 81.19 |
| | BiViT [19] | 1-1 | 9.49 | 0.37 | 59.51 |
| | **SI-BiViT** | **1-1** | **9.87** | **0.45** | **63.80** |
| Nest-T | Full precision | 32-32 | 69.85 | 5.83 | 81.07 |
| | BiViT [19] | 1-1 | 5.00 | 1.22 | 49.89 |
| | **SI-BiViT** | **1-1** | **5.18** | **1.19** | **56.43** |

using Mask R-CNN. Moreover, in the Cascade Mask R-CNN setting, the performance of our SI-BiViT is even more impressive, achieving 30.5% average precision (AP) in object detection and 28.5% AP in instance segmentation. These results further underscore the effectiveness and adaptability of SI-BiViT across various downstream tasks.

## 4.3 Ablation Study

*4.3.1 Overall of SI-BiViT.* Our SI-BiViT primarily comprises two key components: the dual branch and the decoupled training strategy. The dual branch allows us to retain more information from pre-trained models, resulting in performance benefits. As depicted in Table 5, the incorporation of a dual-branch structure contributes to an improvement of 8.35%. To optimize the dual branch more effectively, particularly the SI Module, we introduce the decoupled training strategy, which further enhances performance by a significant margin of 11.7%.

*4.3.2 Effects of Channel-wise Balancing Factor.* To enhance spatial interaction in binarized vision transformers, we utilize a channel-wise balancing factor to integrate the SI module alongside the MLP module. This channel-wise balancing factor aims to strike a balance between these two branches. As shown in Table 6, there is a noticeable improvement after applying these factors, with an increase of 1.83% in ViT-Small and 1.43% in Swin-Tiny. Subsequently, we visualize the channel-wise balancing factor in Figure 5 and observe significant variations across different channels.

*4.3.3 Effects of Token-mixing Linear.* To assess the effectiveness of the token-mixing linear, we compare it with other SI modules, including multi-head self-attention [49] and Cycle-FC [8]. Multi-head self-attention (MSA) enhances spatial interaction through the attention mechanism, necessitating the computation of $Q$, $K$, and $V$, which results in increased computational complexity. On the other

**Table 4: Comparison with State-of-the-art methods on MS-COCO dataset.**

| Framework | Task | Method | Bits$_{(W/A)}$ | AP | AP$_{50}$ | AP$_{75}$ |
|---|---|---|---|---|---|---|
| Mask R-CNN | Object Detection | Full precision | 32-32 | 42.7 | 65.2 | 46.8 |
| | | BiViT [19] | 1-1 | 23.7 | 42.2 | 23.8 |
| | | **SI-BiViT** | **1-1** | **25.3** | **44.1** | **25.6** |
| | Instance Segmentation | Full precision | 32-32 | 39.3 | 62.2 | 42.2 |
| | | BiViT [19] | 1-1 | 23.6 | 40.1 | 24.5 |
| | | **SI-BiViT** | **1-1** | **24.9** | **41.9** | **25.8** |
| Cascade Mask R-CNN | Object Detection | Full precision | 32-32 | 47.0 | 65.7 | 51.1 |
| | | BiViT [19] | 1-1 | 30.0 | 46.8 | 32.2 |
| | | **SI-BiViT** | **1-1** | **30.8** | **47.9** | **32.5** |
| | Instance Segmentation | Full precision | 32-32 | 41.1 | 63.4 | 44.5 |
| | | BiViT [19] | 1-1 | 27.8 | 44.5 | 29.7 |
| | | **SI-BiViT** | **1-1** | **28.5** | **45.6** | **30.0** |

**Table 5: Ablation study of components in SI-BiViT.**

| Method | Top-1(%) |
|---|---|
| Baseline | 38.32 |
| + dual-branch | 46.67$_{(+8.35)}$ |
| + decoupled training strategy | 58.45$_{(+11.78)}$ |

**Table 6: Effect of Channel-wise Balancing Factor(CBF).**

| Model | LSF | Size$_{(MB)}$ | Ops$_{(G)}$ | Top-1$_{(\%)}$ |
|---|---|---|---|---|
| ViT-S | ✗ | 1.84 | 0.21 | 56.62 |
| | ✓ | 1.84 | 0.21 | 58.45$_{(+1.83)}$ |
| Swin-T | ✗ | 7.41 | 0.44 | 62.86 |
| | ✓ | 7.41 | 0.44 | 64.29$_{(+1.43)}$ |

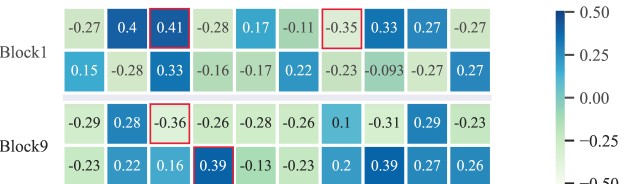

**Figure 5: Visulization of channel-wise balancing factor(CBF). We randomly pick them from block 1 and block 9. The red mark highlights the maximum and minimum elements.**

hand, Cycle-FC augments spatial interaction and performs feature extraction within a token. However, the efficiency of Cycle-FC is not as effective as token-mixing linear in terms of spatial interaction. As demonstrated in Table 7, token-mixing linear outperforms both MSA and Cycle-FC in terms of efficiency and performance.

We can further control the Spatial Interaction Capability (SIC) in the Token-mixing Linear by adjusting the parameter $n'$ in Equation 8. We compare three settings: token-Linear-a, token-Linear-b, and token-Linear-c, which increase spatial interaction by 2 times, 3 times, and 5 times, respectively. As illustrated in Table 8, token-Linear-c achieves the best performance, with a top-1 accuracy of

**Table 7: Comparison with other SI modules.**

| SI Module | Size(MB) | Ops(G) | SIC(times) | Top-1(%) |
|---|---|---|---|---|
| - | 1.79 | 0.21 | 1 | 52.31 |
| Cycle-FC | 1.85 | 0.22 | 1.2 | 52.49 |
| MSA | 1.99 | 0.25 | 2 | 53.06 |
| **token-Linear** | **1.84** | **0.21** | **3** | **58.45** |

**Table 8: Comparison with different Token-Linear settings.**

| SI Module | Size(MB) | Ops(G) | SIC(times) | Top-1(%) |
|---|---|---|---|---|
| - | 1.79 | 0.21 | 1 | 52.31 |
| token-Linear-a | 1.83 | 0.20 | 2 | 57.32 |
| **token-Linear-b** | **1.84** | **0.21** | **3** | **58.45** |
| token-Linear-c | 1.85 | 0.24 | 5 | 58.52 |

**Table 9: Impact of spatial interaction module on different binarized ViT methods on Imagenet. SI module can be directly plugged into prevailing binarization methods.**

| Method | SI Module | Size(MB) | Ops(G) | Top-1(%) |
|---|---|---|---|---|
| BiT[35] | ✗ | 1.65 | 0.21 | 27.90 |
|  | ✓ | 1.84 | 0.21 | 37.15 (+9.25) |
| BiBERT[43] | ✗ | 1.65 | 0.21 | 36.65 |
|  | ✓ | 1.84 | 0.21 | 43.15 (+6.50) |
| BiViT[19] | ✗ | 1.65 | 0.21 | 42.91 |
|  | ✓ | 1.84 | 0.21 | 51.04 (+8.13) |
| SI-BiViT | ✗ | 1.79 | 0.21 | 53.31 |
|  | ✓ | 1.84 | 0.21 | 58.45 (+5.14) |

58.52%. However, this improvement in performance comes at the expense of much more computational cost. On the other hand, token-Linear-b strikes a balance between performance and efficiency, achieving a top-1 accuracy of 58.45%.

*4.3.4 Combination with existing Binarized ViT methods.* Our SI module is orthogonal to existing Binarized ViT approaches, allowing it to be seamlessly integrated into existing binarization methods to further enhance performance. We apply the SI module to four prevalent binarization methods: BiT[35], BiViT[19], BiBERT[43], and the baseline model. As shown in Table 9, we observe an improvement of 8.13% in BiViT, with an average increase of 6.59% across all methods. Our SI module significantly enhances the performance of all four methods, underscoring the strong flexibility and effectiveness of SI-BiViT. Experiments on Tiny-ImageNet can be found in Appendix.

*4.3.5 Effects in Full-Precision Backbones.* As our SI-BiViT introduces a new dual branch and modifies the structure of ViT, it is necessary to explore the impact of this modification on the full-precision model to ensure whether the improvement is introduced by improved full-precision models. We train the full-precision ViTs with the proposed dual branch and the results are shown in Table

**Table 10: Comparision of SI module on full-precision and binarized models.**

| Model | Bits(W/A) | SI Module | Top-1(%) |
|---|---|---|---|
| Vit-S | 32-32 | ✗ | 80.14 |
|  |  | ✓ | 79.79 (−0.35) |
|  | 1-1 | ✗ | 53.31 |
|  |  | ✓ | 58.45 (+5.14) |

**Table 11: Comparison of Latency in deployment.**

| Method | Bits(W/A) | Size(MB) | Ops(G) | Latency(ms) |
|---|---|---|---|---|
| Full precision | 32-32 | 87.90 | 4.25 | 2.55 |
| SI-BiViT | 1-1 | 3.07 | 0.21 | 1.12 |

10. In full-precision models, there are negligible performance improvements while there is a significant performance improvement in the binarized model, with 5.69% on ViT and 6.14% on Swin. It demonstrates that the huge performance benefit is brought the increased spatial interaction.

*4.3.6 Latency Comparision.* To measure real-time latency on hardware, we conducted experiments on our binarized model and full-precision model using an Nvidia-3090 GPU. The full-precision matrix operation was implemented using cuDNN[10], while the binarized matrix operation was implemented using TC-BNN [25]. As presented in Table 11, our method achieves a 2.28x reduction in latency compared to its full-precision counterpart. We anticipate that further acceleration could be achieved with the development of hardware specifically designed for binarization acceleration.

## 5 CONCLUSION

In this paper, we reveal that the lack of token-level correlation greatly affects the performance of binarized vision transformers. To measure token-level correlation in different modules, we define spatial interaction capability (SIC) and observe that existing MLP modules only extract features in each token independently, which severely hinder spatial interaction in binarized ViTs. This motivates us to propose SI-BiViT. Our proposed method consists of an SI module, which is placed alongside the MLP module to formulate a dual-branch structure. This structure not only leverages knowledge from pre-trained models but also increases spatial interaction through the SI module. To avoid gradient conflicting problems, we also design a decoupled training strategy to train these two branches more effectively. Importantly, our SI-BiViT is orthogonal to existing Binarized ViT approaches and can be seamlessly integrated to further improve their performance. We conduct extensive experiments that demonstrate the flexibility and effectiveness of SI-BiViT across four classic ViT backbones when applied to several downstream tasks.

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
