# OpenReview forum: "SI-BiViT: Binarizing Vision Transformers with Spatial Interaction"
_acmmm.org/ACMMM/2024/Conference — MM2024 Poster_

### Official Review · Reviewer_HmTw · 2024-05-24

**Rating:** 5
**Confidence:** 2

**Summary:**

This paper reveals that token-level correlation is crucial for the performance of binarized vision transformers. The authors propose SI-BiViT to incorporate spatial interaction into the binarization process. They conduct extensive experiments on various downstream tasks to demonstrate the flexibility and effectiveness of their proposed method.

**Strengths:**

1. The paper is well-written and easy to understand.

2. They reveal the importance of token-level correlation in binarized ViTs.

3. The experiments are extensive, and the proposed method achieves promising performance in image classification, object detection, and segmentation tasks.

**Limitations:**

It seems the proposed method shows a more significant improvement in classification performance compared to detection and segmentation. Providing a reasoned explanation for why it is more effective in classification would be beneficial and enhance understanding.

**Suitability:**

3

---

### Official Review · Reviewer_7w2L · 2024-05-24

**Rating:** 4
**Confidence:** 3

**Summary:**

The authors introduce a method termed SI-BiViT, which integrates Spatial Interaction into the binarization procedure. Specifically, an SI module is incorporated alongside the MLP module, establishing a dual-branch architecture. This design not only harnesses knowledge from pretrained ViTs by distillation through the original MLP but also bolsters spatial interaction via the inclusion of the SI module. Accompanying this, a decoupled training strategy is devised to facilitate more effective training of these dual branches. Crucially, the proposed SI-BiViT approach is orthogonal to existing binarization techniques for ViTs and can be seamlessly integrated.

**Strengths:**

1. Well-written and easy to understand in most parts.
2. The experiments show promising results across multiple datasets.
3. For the first time, it proposes and quantifies the lack of "Spatial Interaction Capability" in Binarized Vision Transformers (BiViTs), highlighting its significance for performance.

**Limitations:**

1. Perhaps some demonstration of visual effects could be shown.
2. In Tables 2 and 3, there appears to be an error in the application of bold formatting. Specifically, numbers that do not surpass the SOTA values have also been incorrectly bolded.
3. The decline in performance when using full precision with SI deserves further explanation and more comprehensive experimentation.

**Suitability:**

3

---

### Official Review · Reviewer_3M9G · 2024-05-25

**Rating:** 5
**Confidence:** 3

**Summary:**

This paper aims to reduce the performance gap between binarized and full-precision ViT models. This paper first empirically demonstrates the lack of token-level correlation severely limits the performance of binarized ViTs. So this paper proposes SI-BiViT, which integrates spatial interaction into the binarization process. Comprehensive experiments show the effectiveness and flexibility of the proposed model.

**Strengths:**

1. The authors first employ an empirical study of spatial interaction in binarized ViTs, showing that the lack of token-level correlation results in a zero spatial interaction capability and further limits the binarized ViT performance.

2. This paper proposes a dual-branch structure by incorporating a Spatial Interaction Module alongside the existing MLP module to integrate spatial interaction into the binarization process. This paper also designs a decoupled training strategy to effectively optimize the dual branch.

3. The proposed SI-BiViT is effective on various ViT backbones and downstream tasks. Also, it is flexible to plug into ViT backbones.

**Limitations:**

1. The authors introduce the concept of SIC to measure token-level correlation in binarized ViTs. The authors do not clarify whether there are existing measurements for this. If yes, what is the difference between the proposed one and the existing one? If not, what are the motivations for proposing SIC, and why this can correctly measure the correlation?

**Suitability:**

2

---

### Meta-Review · Area_Chair_Ah7C · 2024-07-07

**Recommendation:** Accept (Poster)
**Confidence:** 5

**Metareview:**

Overall, the reviewers all express opinions of the acceptance of this work. The paper is deemed moderately to definitely suitable for the multimedia/multimodal processing community, with all reviewers confident in their assessments. The final decision tends towards acceptance, influenced by the author's rebuttal and additional clarifications provided during the review process. After collecting all the comments and discussion, AC decided to accept this paper as a Poster.

Pls further polish the paper according to the rebuttal and comments. Look forward to seeing the final version.

---

### Meta-Review · Senior_Area_Chairs · 2024-07-10

**Recommendation:** Accept (Poster)
**Confidence:** 5

**Metareview:**

All the reviewers gave positive ratings and tend to accept the paper. SAC and AC agree with reviewers and recommend acceptance of the paper.